# Collaborative Sensing with Interactive Learning using Dynamic Intelligent Virtual Sensors

**DOI:** 10.3390/s19030477

**Published:** 2019-01-24

**Authors:** Agnes Tegen, Paul Davidsson, Radu-Casian Mihailescu, Jan A. Persson

**Affiliations:** Internet of Things and People Research Center, Department of Computer Science and Media Technology, Malmö University, 20506 Malmö, Sweden; agnes.tegen@mau.se (A.T.); paul.davidsson@mau.se (P.D.); radu.c.mihailescu@mau.se (R.-C.M.)

**Keywords:** virtual sensors, sensor fusion, machine learning, dynamic environments, Internet of Things

## Abstract

Although the availability of sensor data is becoming prevalent across many domains, it still remains a challenge to make sense of the sensor data in an efficient and effective manner in order to provide users with relevant services. The concept of virtual sensors provides a step towards this goal, however they are often used to denote homogeneous types of data, generally retrieved from a predetermined group of sensors. The DIVS (Dynamic Intelligent Virtual Sensors) concept was introduced in previous work to extend and generalize the notion of a virtual sensor to a dynamic setting with heterogenous sensors. This paper introduces a refined version of the DIVS concept by integrating an interactive machine learning mechanism, which enables the system to take input from both the user and the physical world. The paper empirically validates some of the properties of the DIVS concept. In particular, we are concerned with the distribution of different budget allocations for labelled data, as well as proactive labelling user strategies. We report on results suggesting that a relatively good accuracy can be achieved despite a limited budget in an environment with dynamic sensor availability, while proactive labeling ensures further improvements in performance.

## 1. Introduction

With the evolution of sensor technology, Internet of Things (IoT), and high performance communication networks, making use of the vast amount of data being generated is critical for developing new and more powerful applications. This may concern supporting the user by distilling information from the incoming sensor data streams and presenting the user with the relevant sensory data just-in-time, while accounting for the user’s interaction with the cyber-physical systems in its environment. It may also concern how the sensor data produced by a specific device can be used concurrently in multiple applications. One way to address these new challenges is to create an abstraction layer, which is overlaying the physical infrastructure. This is often referred to in the literature as a “virtual sensor”, which has the role of isolating applications from the hardware by emulating the physical sensor in software [1].

The abstraction layer provided by a virtual sensor can cater for multiple logical instances, which may support various applications with various goals, where the different virtual sensors may make use of overlapping sets of sensors [2]. Hence, it can enable a flexible way to develop services based on fusing sensor data either owned by the same provider or obtained from multiple sensor infrastructure providers. Overall, virtual sensors may facilitate concurrency by means of having different, possibly overlapping, subsets of sensors committed to different tasks. A key aspect of virtual sensors, which we are addressing in this work, has to do with measuring properties for which no corresponding physical sensor exist, e.g., the type of activity that is going on in a room. That implies that the virtual sensor needs to be capable of data fusion.

### 1.1. Related Work

An example of deploying virtual sensors is in handling the situation when some sensors are malfunctioning. Such a solution is proposed in [3], where the authors present VirtuS, a prototype implementation of virtual sensor for TinyOS. The system uses stacks to store data temporarily, offering the possibility to perform basic arithmetic operations, e.g., computing the mean value of a number of data readings, before returning the value to the application. 

Another use of virtual sensors is that of removing the occurrence of outliers in sensor readings. It is typically based on correlating data from a number of the same type of sensors, located in proximity to each other. As a result of this aggregation, not only are the readings more reliable but also, transmission is more efficient since less data needs be transmitted to the upper layers. This is particularly important in wireless sensor networks [4]. Similarly, in [5] a virtual sensor approach is employed to gather track-data from several visual sensors and to determine a performance score for each sensor based on the coherence with the rest of the data. This information is then fed back to the sensor, allowing it to use it as external information in order to reason about the confidence of its readings. Along the same lines, active noise control is an often addressed topic in the context of virtual sensing. The challenge here is to create quiet zones around certain target points, without actually placing sensors at those desired virtual locations [6]. Among other applications, we also note the traffic domain where virtual sensors are used to provide congestion information in real-time, such as in the case of Washington state’s traffic system [7]. 

A common characteristic for many of the existing virtual sensor solutions is that they provide a way to process homogeneous types of data, generally retrieved from a predetermined grouping of sensors. However, there are also a number of examples where virtual sensors combine heterogeneous data types for computing new values. Mobile devices, such as smartphones are a clear example of exploiting the sensors embedded in the device: accelerometers, gyroscope, GPS, microphone, camera, proximity sensors, ambient light sensors, and so forth, in order to fuse the data and obtain an overall and more complete perspective of the environment and the users’ activities [8]. Note that in such an application the set of sensors are constant and are normally assumed to always be available. 

### 1.2. Contribution

In this paper, we introduce a refined version of the Dynamic Intelligent Virtual Sensors (DIVS) concept [9] and validate some of its properties by experiments. A DIVS can be seen as a logical entity that takes the output from physical sensors and produces novel data that is needed by the applications/services. One main refinement of the original concept is to include interactive machine learning, by introducing the user-in-the-loop in order to improve accuracy and minimize the requirement for labelled data.

In cooperation with companies and facility management representatives we have identified a number of potential services for users including: understanding the usage of multipurpose office environments, guidance for users of the building and real-time monitoring of the number of people at different areas of an office complex, e.g., for evacuation. DIVS is intended to directly support such services through the possibility to define the suitable property of the environment, which it should be monitoring, as well as improving its accuracy by utilizing feedback regarding its output.

The paper is organized as follows: in Section 2 we propose a refined version of the DIVS concept. In Section 3 we demonstrate and evaluate our proposed approach in the context of a typical IoT setup. In Section 4 we discuss results and Section 5 concludes the paper. 

## 2. Materials and Methods

A DIVS produces virtual sensor data for which there is typically no off-the shelf sensor available (or difficult/expensive to develop). For instance, it may produce sensor data, which is a classification of the states of an environment by measuring a multitude of physical properties through physical sensors. The DIVS concept is broader than that of a virtual sensor per se, which is typically used in devices with a fixed set of sensors, such as in smartphones. In contrast, in our previous work [9] we introduced DIVS to denote an abstract measurement resulted from combining a number of heterogeneous data sources provided by a group of physical sensors and/or other DIVS, whereas the set of sensors supporting a DIVS can be adapted over time to better accommodate the current context and accuracy of the measurement. In real-world distributed sensing there are numerous instances where such adaptation needs to be performed on-the-fly in order to cope with the inherent uncertainty of the sensor infrastructure, such as sensor malfunctions, mobile sensors that are entering or leaving the system, new (type of) sensors being appended to the system, sleep cycles of sensors, battery constraints etc.

In addition to the aforementioned challenges, in this paper we extend the DIVS concept by focusing on some of the key learning modalities responsible for producing the DIVS output, in response to several issues that are characteristic to the IoT domain. In particular, we aim at addressing the inadequacy of standard machine learning approaches in computing the prediction (estimation or classification) function of the DIVS in situations when:there are changes to the patterns observed in the sensor data are occurring over time, cf. concept drift [10];sensor data streams suddenly disappear, or new sensor data streams appear;there are a lack of labeled data available for supervised learning, at least initially, cf. cold start problem [11] and one-shot learning [12];there is a need to efficiently and rapidly configure the DIVS;it is not possible to store all sensor data and therefore incremental learning [13] must be used;the ability of reusing information learned in one context to another context is desired, cf. transfer learning [14].

Our main goal is to propose a general-purpose design for virtual sensing with high versatility across different environments and applications. To this end, we start by revising the set of DIVS attributes in order to ensure the minimum viable data quality characteristics that enables us to tackle these challenges.

### 2.1. Data Quality Characteristics

Data obtained from IoT sensors and devices comes in various types and formats, is often noisy, incomplete and especially inconsistent [15]. Even when standardized ways are used to connect sensors to IoT platforms, such as using the MQTT or REST protocols or the JSON format, the data “packaging” is often carried out in unconventional ways, leading to data descriptions that are only human-interpretable. Importantly, data quality metrics can assess the usefulness of the data for a data consumer. In [16] the authors provide a study of data quality, while the work of [17] discusses data quality in IoT, emphasizing its importance in order to deliver user engagement and acceptance of the IoT paradigm and services. Contrary to common belief, data accuracy is just one of the aspects that ensure high qualitative data. According to [16], data quality can be evaluated in terms of the following dimensions and metrics respectively i) intrinsic: accuracy and reputation, ii) contextual: timeliness, completeness and data volume, iii) representational: interpretability and ease of understanding, iv) accessibility: accessibility and access security. In the following, we introduce a refined version of the DIVS attributes in response to the new set of challenges presented in the beginning of Section 2, while accounting for the data quality metrics previously identified. In order to ensure a minimal representation of the DIVS output, we introduce the distinction between mandatory and optional data attributes, as well as, data items that are continuously streamed and data items that are only transmitted when needed or made available upon request. The continuously streamed data items are:Sensor ID (iii)Timestamp (ii)Sensor value (i)

Whereas the data items provided on request or when needed are:Data type (iii)Property type (iii)Range of values (iii)Unit of measurement (iii)Physical location being sensed (ii)Estimated accuracy of measurement (i)—optionalEvent triggered or periodical sensing (iv)—optionalInformation gain (ii)—optional

The attributes of DIVS, which represent in our view the basic metadata description of a virtual sensor, can be directly correlated to the data quality dimensions in [16]. For instance, the contextual dimension is given here through the timestamp and physical location attributes, which capture the spatio-temporal characteristics of the data. The physical location being sensed is especially important for the case of mobile sensors and location should in such cases be transmitted when changed. In addition, the information gain holds further contextual information about sensors data consumed by the DIVS (see Section 2.2). Representational information, which ensures the interpretability of the data, is given via the sensor ID, data type, property type (what aspect of the environment is being sensed), range of values and unit of measurement. Except for providing a basic characterization of the data, this information can further provide input for the pre-processing phase, such as detecting malfunctioning sensors that are producing data values that mismatch the given data type or are out of range. Moreover, the range of values attribute is essential, whenever a normalization or scaling of the data needs to be performed. A special attribute, related to accessibility, is assigned for specifying whether the sensor data is streamed at periodical intervals of time or the transmission is event triggered. The intrinsic dimension in terms of data quality, is given here by the estimated accuracy of measurement, which is a self-assessment, confidence measure of the DIVS about its prediction, or a precision of measurement in the case of regular sensors. 

In Figure 1 we outline the system architecture of DIVS, emphasizing the functionalities of the core components. One main refinement of the original DIVS concept is to include the feedback feature, which enables DIVS to actively query the user or other DIVS regarding the accuracy of the data. Thus, we provide a real-time architecture capable to integrate feedback information on-the-fly in order to improve performance. In the following, we provide a functional description of each of the DIVS components.

### 2.2. Dynamic Sensor Selection

In terms of data volume, IoT sensors can be regarded as one of the main data producers. However, the proliferation of sensing devices through large-scale deployments often comes at a cost in terms or sensor reliability (i.e., uncertainty of sensor availability) and noisy data. Moreover, sensors may be mobile, e.g., embedded in a smartphone, which adds to the uncertainty of the availability. Under these circumstances, it becomes imperative to provide DIVS with built-in mechanisms to assess which sensors are most significant in relation to the predictive task at hand. This problem is exacerbated when limited labelled data is available for training the model, in which case, a reduced number of relevant sensors is preferred. 

Given a set of sensors representing the available data sources for a DIVS, the DIVS keeps track of a dynamic metric, termed information gain (IG). The IG has the role of assessing the usefulness, or the importance of a particular sensor for computing the desired output of the DIVS. There are clearly various ways for specifying the IG. For instance, in information theory terms, the notion of information gain is used to denote the reduction in the uncertainty of a random variable, given the value of another variable [18]. Important to note is the fact that here we use IG not merely as static parameter computed at design-time, but one that needs to be continuously updated based on the data provided by the sensor at run-time. In addition, we may introduce a threshold value, such that a sensor is not considered by the DIVS whenever its IG drops below the threshold. Furthermore, the use of IG can be instrumental for scenarios where some form of transfer learning is required. Consider for instance the case of a DIVS monitoring the type of activity in an office space. Suppose now that we want to run the same DIVS to monitor a new area of the building based on a slightly modified mix of heterogeneous sensors. Thus, having a good understanding of the usefulness of a particular type of sensor input for computing the DIVS output, enables us to reuse information learned in one context to another context. 

### 2.3. Online Learning

The standard scenario for machine learning is the so-called batch learning or offline learning, where a model is trained on a given set of labelled instances. We have considered this approach for DIVS in previous work [9], where heterogeneous sensor data undergoes a data fusion and classification procedure via batch learning, in order to produce the DIVS output, as depicted in Figure 1. In contrast, in incremental learning a model is trained on data which arrives over time in streams. Online learning is a specific type of incremental learning where, after every time step, an updated model is computed based on the previous model and the new data point (or mini-batch of data points) [13]. Most methods used for online learning are standard Machine Learning approaches that has been adapted for an online setting. For instance, online versions of Support Vector Machines, Random Forest and Naïve Bayes classifier has been suggested [19].

In this work, we augment DIVS with online learning capabilities in order to cope with two types of situations that are prevalent in the IoT ecosystems. In the first type of scenario the aim is to improve efficiency and scalability of an already existing model that has been trained in the traditional batch learning fashion with a labelled dataset. The second type of scenario is to handle situations where data naturally arrives over time and no labelled data is available in advance. Especially in the case of virtual sensing, it is key that the system is able to display online learning capabilities. That is, on the one hand, to continuously update an initially deployed model such that it is adaptable to the possibly changing target concept [20], while on the other hand, it can deal with the cold start problem when needed. This functionality provides DIVS with the versatility to operate in settings where the system can either leverage on historical data, or kick-start its process without the assumptions of previously labelled data. Moreover, given the dynamic nature of IoT, an online learning model can quickly adapt to incorporate new types of features (i.e., new sensors), or to discard them. It is also generally the case that online learners do not require to store the incoming data, which makes them a suitable candidate for pervasive embedded devices. 

### 2.4. Active Learning with User-in-the-Loop

We further extend the DIVS concept by integrating an active learning mechanism. This assumes that DIVS are able to receive a certain level of user feedback or possibly, feedback from other DIVS. Especially in the situation of a cold start, unless, the learning algorithm used is unsupervised, it is critical to have a way of interacting with the users of the system in order to obtain labelled data. However, labelling data is in often expensive and might in many cases be difficult to obtain. A solution to this problem could be to only label selected instances, instead of all data. By allowing the learning method to choose which data points to label, the number of labeled instances can decrease, while still matching, or possibly even exceeding, the performance of a learner trained on a fully annotated dataset. Letting the learner decide what data points to learn from is referred to as Active Learning. The Active Learning method does this by posing a query to an “oracle”, which then provides a label for the given data point. The “oracle” can be a human user or expert, or it can be another system. How many instances can be queried is defined by a labelling budget. The selection process of which instances to query can be made based on different criteria depending on the problem at hand. A common selection criterion is for the learner to query for the instances where it is least certain [21].

Recall now that for our IoT setup we are dealing with streaming sensor data, which means that the query decisions must be made on-the-fly, as data arrives. This poses more demands on the learning model than if all data is readily available. The labeling budget cannot be calculated over an infinite time horizon, which means that the budget must be balanced throughout the learning process. Different approaches to tackle these problems are presented and on compared in [22].

To sum up, the architecture of the DIVS data pipeline is shown in Figure 1. The four major processes are as follows: (1) Data pre-processing, which has the role of preparing the incoming sensor data for further processing by smoothing noisy data, identifying and removing outliers, and resolving data inconsistencies; (2) Dynamic sensor selection and featurization, which makes use of information gain in order to identify the most suitable sensors, possibly reducing the dimensionality of the data and/or introducing additional statistical features (e.g., variance, FFT, etc.), as well as allowing DIVS to operate in a dynamic environment; (3) Online Incremental Learner, which is responsible for ensuring an open-ended learning procedure, capable to incorporate new data in order to improve the DIVS performance; (4) Active Learning, which fosters the interaction with users or other DIVS, by allowing it to obtain labelled data, while at the same time minimizing the amount of queries to the user.

## 3. Results

To illustrate the DIVS concept and validate some of its properties, we present a set of experiments where a DIVS uses data streams from a heterogeneous set of sensors to estimate the occupancy of an office room. Sensors might physically leave or enter the space, they might stop working or a new sensor might be added. This results in the availability of the sensors being dynamic, i.e., the set of data streams delivered to the DIVS is not constant over time. Thus, the DIVS must continuously adapt to make best use of the data from the sensors that are currently available.

As is the case in many real world applications, we assume that there is no labelled data from the given environment available prior to the deployment of the DIVS, corresponding to a “cold start”. Data from the currently active sensors arrives gradually over time, but to also obtain corresponding labels, i.e., whether the office space is occupied or not, feedback from a user is needed. In some of our scenarios, it is possible for the DIVS to query users for labels, but also for the users to proactively provide labels when they consider it necessary. Different active learning strategies, as well as strategies with a proactive user, are presented in the experiments described below.

### 3.1. Dataset

We use the collection of datasets from [23]. The data collection is made up of three different datasets using the same set-up in a room in an office setting, but over different periods of time. Each dataset contains a recorded sequence of one data instance per minute from five features (light, temperature, humidity, CO_2_-level and humidity ratio) along with whether the room was occupied or not at that point in time. The length of the data streams is between 2664 instances (1.8 days) and 9752 instances (6.8 days). The state of the room is occupied during similar hours and in similar patterns to what a regular work day might look like. On most work days it is occupied from around 8:30 a.m. to 5:30 p.m., with a longer period of being unoccupied around lunch time and some shorter periods throughout the day. During days corresponding to the weekend however, the room was unoccupied for the entire day.

In the experiments, the dynamic availability of the sensors is simulated by restraining the access to data from some of the sensors during periods of time. We randomly generate from which sensors data is withheld, when it is withheld and for how long. This represents the mobility of the sensors, i.e., when sensors leave the environment and therefore stop streaming data or new sensors appear and start to stream data. While the sensors could stop uploading due to e.g., sensors breaking down or network failures, the longest dataset is less than a week long and contains five features. The probability in a similar real world scenario of one or several of the sensors, to stop uploading for these reasons within a week is not very high. Therefore, we did not base the simulation of a dynamic setting on mean time between failures or equivalent. Instead, we randomly generated each scenario as follows. One to three sensors, out of five, were randomly chosen for each simulation. For each of these sensors it was randomly generated whether they would be disappearing sometime during the sequence (i.e., stop uploading data) and/or appearing (i.e., going from not uploading data, to start uploading data) and at which point in time this would happen. This means that a sensor can disappear and stop streaming data at one point, but later reappear and start streaming again, during the same simulation. Thus, within each scenario at each point in time at least two sensors are uploading data, but during periods of time at least one sensor is absent, to represent the mobility of the sensors.

### 3.2. Learning Method

We use a weighted Naïve Bayes approach for classification and the Variable Uncertainty Strategy for active learning. Naïve Bayes classifier has several advantages given the type of setting we are looking at. It requires fewer training examples compared to many other machine learning algorithms [24,25] and is suitable for online learning, i.e., when one instance of data is processed at a time. The classifier is not computationally complex, which is of importance when producing real-time predictions from streaming data. This makes it a popular choice for when annotated data is scarce, but also when the dataset is gradually built up [26].

The set of sensors supporting a DIVS can change over time, which means that a learning model that can handle this dynamic setting is needed. The different features used for classification in Naïve Bayes are handled separately, which makes it possible to add a new one (if a new sensor appears) or discard an old one (if the sensor stops uploading data), without retraining the entire model. 

In DIVS, information gain is used to describe the explanatory power of one property towards computing the output value. In a classic Naïve Bayes classifier, each feature is equally influential on the output. With information gain however, the influence on the output of each separate feature can be adjusted. The information gain is represented by weights which are assigned to each feature of the model. The weights are calculated from the overlapping coefficient [27,28] of each feature and its distributions for the separate class labels.

In [22], Variable Uncertainty Strategy was suggested as a well-performing active learning approach for data streams when no strong concept drift is expressed. The approach aims to label the instances which it is least certain about, within a given a time interval and labelling budget. The budget is expressed as a ratio of how many of the incoming data points can be queried compared to all incoming data points. Since it is not possible to calculate a ratio for the future incoming instances of the data stream, an estimation is made based on a sliding window. The sliding window contains information regarding which of the latest incoming data points, within the specified time interval, has been queried. Furthermore, since in many real world applications is not possible to store all values from the data streams, only a limited number of labelled instances for each class is stored to be used for the Naïve Bayes classifier. These are also selected based on a sliding window approach, i.e., if the maximum number of stored instances allowed is reached and a new labelled data point arrives, the oldest instance gets discarded, something which also may support the handling of potential concept drift.

### 3.3. Experimental Setup

The experimental setup was designed to illustrate some of the core properties of the DIVS concept. In all of the experiments, the DIVS starts without any labelled data, i.e., cold start, and must gradually obtain this from a user. Different learning approaches were tested in a dynamic setting regarding the set of sensors streaming data, to demonstrate how a DIVS handles a changing set of input values. To illustrate how different types of user feedback can influence the performance, we tested different strategies for the user to provide input, from taking the role of “oracle” (as in classic Active Learning) to being more proactive (as in Machine Teaching [29]). We also show how different budget ratios for querying, i.e., how much the DIVS can ask for feedback from a user affects the accuracy of classification. To get variation in the sequence of incoming data, the data was divided by days (from midnight to midnight) and randomly shuffled for each simulation.

If nothing else is specified, the number of labelled instances stored for each class is set to 50, i.e., 50 labelled data points from when the room was occupied and 50 from when it was not. Tests were carried out where the number of stored labelled instances was increased, but this did not improve the accuracy significantly. 

A limitation of the Naïve Bayes classifier is that at least two labelled data points for a class are needed for it to be able to calculate a standard deviation. However, the learner should produce predictions from the start. Thus, before the Naïve Bayes classifier can be used to make predictions, the label of the last query is used for prediction.

### 3.4. Budget Ratio vs Accuracy

To illustrate how changing the labelling budget affects the accuracy, different budget ratios were tested with the Variable Uncertainty Strategy. Each budget ratio was used for 50 simulations of shuffled days to produce an average of the accumulated accuracy at the latest point in time and its standard deviation. For this experiment all sensor values were included at all times. 

Figure 2 displays the accumulated accuracy for different labelling budgets. With an increased labelling budget, the accuracy increases, as would be expected. This is most significant when the labelling budget is low, in which case increasing it boosts the performance significantly. Also, note that even with a 0.2% labelling budget, the leftmost result in Figure 2, the performance is above 75% on average, while 95% accuracy is achieved with a 20% labelling budget.

### 3.5. Learning Strategies in Settings with Static Set of Sensors

In this experiment we test how incremental learning with an active learning approach performs compared to an iterative batch learning approach. A static setting regarding the presence of sensor data was used, i.e., all sensor values were accessible at all times. The learning strategies were tested on 50 simulations of shuffled days, to produce an average accumulated accuracy over time.

Iterative Batch Learning: This learning strategy has unlimited budget and obtains labels for all incoming data points. It also has no restriction on how many labelled instances can be stored. Unlike the other approaches described below, that are all versions of incremental learning, this can be compared to doing batch learning with all thus far encountered data instances for each point in time. This strategy is used as a baseline, as there are no restrictions on labelling budget or storage of data.

Variable Uncertainty Strategy: The core of this active learning approach is described in Section 3.2 Learning Method. For this experiment a labelling budget of 1% was used, which was based on giving good enough accuracy, while not bothering a user with too many queries. 

Figure 3 shows how the active learning approach compares to Iterative Batch Learning. Iterative Batch Learning performs well from the start, after 1 h the accumulated accuracy is 97.80% and after 1 h it keeps increasing. This is expected, because of the unlimited labelling budget and unlimited memory of the learning strategy. The accumulated accuracy of the Variable Uncertainty Strategy does not perform as well as the baseline. Yet, after 144 h (or 6 days), the accumulated accuracy reaches almost 91%, while only using 1% of the labelled instances that the Iterative Batch Learning uses. 

Table 1 shows the F1 scores for the two strategies after 144 h. Here, the Iterative Batch Learning outperforms the Variable Uncertainty Strategy, again, as expected. Still, the F1 score of the latter strategy is 0.8111, as compared to the F1 score of 0.9698 for the baseline strategy.

### 3.6. Learning Strategies in Settings with Dynamic Set of Sensors

This experiment compares how different learning strategies perform in a dynamic sensor setting. In total, 100 simulations of randomized dynamic settings and shuffled days were made, to generate an average of the accumulated accuracy over time:

Variable Uncertainty Strategy: The method with the same parameter settings as described above was used.

Random Strategy: This approach has the same budget as the Variable Uncertainty Strategy, but instead of querying based on whether the certainty is low, the learner queries randomly.

Decreasing Budget Strategy: In this strategy the learner has unlimited budget for querying for a short period at the start, until it has collected 50 labelled data points for each class. After this, the labelling budget decreases to 0.5% for the rest of the time. This could be seen as a user needing to constantly provide labels in the beginning, but after this is done, the user is questioned less. The total amount of queries over time is parallel to the other approaches.

Figure 4 displays the accumulated accuracy for the different learning strategies over time. The Random Strategy performs worse than the others at the beginning, but catches up to and even surpasses the Variable Uncertainty Strategy and performs on the same level thereafter. The performance of a Naïve Bayes classifier is dependent on having labelled data that sufficiently represents the current distributions of the output classes. When enough instances have been collected, and given that the set is continually updated, the models for the two different strategies becomes gradually more similar. Another reason for the Random Strategy improving in relation to the Variable Uncertainty Strategy is that, especially after a while, the latter approach might not be as good at estimating its own accuracy. From the F1 scores displayed in Table 2, it is also clear that the Variable Uncertainty Strategy and the Random Strategy achieve a similar accuracy, while the Decreasing Budget Strategy does not perform as well.

### 3.7. User Strategies in Settings with Dynamic Set of Sensors

To examine how different approaches of a user providing feedback can affect the accuracy, strategies where the user was proactive to different degrees were tested in a dynamic sensor setting. As in the previously described experiment, 100 simulations were generated and an average of the accumulated accuracy over time was calculated from this:

Variable Uncertainty Strategy with oracle: The method with the same parameter settings as described above was used. Whenever the learner queried the user, the user provides a label. 

Variable Uncertainty Strategy with proactive user: This approach is similar to one described above, with the difference that the user only answers a query if the latest prediction were wrong.

Proactive user with budget: With this strategy the learner never queries for labels, instead they are provided by the initiative of a user. When the prediction made by the classifier is wrong, the user provides a label given a labelling budget. The labelling budget is the same as for the other approaches.

Proactive user with unlimited budget: In this approach the labels are also provided solely by the initiative of a proactive user, but with an unlimited labelling budget. This means that a new labelled data point is provided each time the learner predicts wrong.

Figure 5 compares different user strategies in a dynamic setting. As expected, a user that has an unlimited budget to correct the learner when it makes a wrong prediction, performs well from the start. The Variable Uncertainty strategy, i.e., letting the learner completely decide when to query, and the strategy where a proactive user provides a label whenever there is a wrong prediction, both given a budget, performs similarly most of the time. By combining these two approaches, that is, the Variable Uncertainty Strategy with a proactive user, the accuracy improves significantly. Interestingly, this does in fact mean that the user overall will provide less labelled data points, compared to the regular Variable Uncertainty Strategy. By not responding to queries when the predictions were correct, but the learner was still uncertain enough to pose the query, the budget can instead be used for the instances where the prediction was not correct, which could explain the increase in accuracy. The F1 scores in Table 3 showcase that all strategies with a proactive user has better accuracy, compared to the one without. The strategy were active learning and a proactive user are combined, however, does again achieve the highest accuracy, out of the strategies with a limited labelling budget.

It is worth noting that while the Variable Uncertainty Strategy used up its labelling budget of 1%, the other two approaches with budgets did not. The strategy with a proactive user used up 42% of the budget (0.42% of all instances) and the Variable Uncertainty Strategy with a proactive user only used up 28% of the budget (0.28% of all instances). The strategy with a proactive user with unlimited budget on the other hand labelled 9.6% of all instances on average. After 144 h (or 6 days), the Variable Uncertainty Strategy with a proactive user had an accumulated accuracy of 92.69%. This can be compared to the approach with an unlimited budget that had an accumulated accuracy of 97.29% after the same amount of time.

Figure 3, Figure 4 and Figure 5 all contain learning curves where the accumulated accuracy is very high in the beginning, then rapidly decreases for a while, after which it increases again, but at a slow and steady pace. This is due to the fact that the predictions at the beginning are strictly based on the label from the latest query. Since the state of the room is not rapidly changing most of the time, this will give a high accuracy for a short while. When the Naïve Bayes classifier has collected at least two labelled instances, it can start to be used to make the predictions. 

## 4. Discussion

The Naïve Bayes classifier was chosen because of its ability to perform well given few training examples (compared to many other Machine Learning methods), its adaptiveness to a changing set of input values and suitability for online learning, among other things. Despite several desirable properties of the method, it also has its limitations. The Naïve Bayes classifier is built on the assumption that all features used are independent from each other, given the output. With datasets from the real world however, this is rarely the case. The assumption of independence among all features does not hold true for the dataset we used either. Still, even though the assumption is violated, the classifier has been shown to perform well in many cases [30]. Another disadvantage with the method is concerning the probability estimations for each output class. These tend to be either very close to 0 or very close to 1. This is not necessarily a problem for the classification, as the classifier only has to choose the class with the highest probability, but it may not give a good estimation of the actual probability. Since the probability is used in the Variable Uncertainty Strategy, this may affect the performance of the active learning. In the future we plan to compare the performance of the Naïve Bayes classifier to other methods with regards to DIVS.

The dataset that all the experiments were tested on was chosen because it is a set labelled with regards to the state on an environment, and which contains a set of heterogeneous features collected as sequences of data. There are however limitations with the chosen dataset. The different states were only two in number, displaying whether the room where the data was collected was occupied or not. Furthermore, the transition from one state to the other is relatively infrequent. Most days the pattern of whether the room was occupied or not followed standard working hours in an office, but for some days the room was unoccupied during the entire day. This means that for long periods of time the state does not change. At the same time, the data is based on a real-life scenario and displays how the proposed approach would perform in a similar setting.

Since the dataset only contains five different features, it was not possible to simulate the type of sensor-intensive scenarios where the dynamic properties of a DIVS could be fully explored. Moreover, even though the features were all different, the dataset only contains features with numerical values. This simplifies the problem of heterogeneous input data, compared to if there for instance had been a mixture of numerical and categorical features for the DIVS to handle. In the future we plan to do experiments with more complex datasets, both regarding the classes and the features.

Another essential property of the DIVS that we intend to explore in future work, is information gain and how it can be used through transfer learning. For example, if a DIVS is trained on a set of sensors in a given environment, but then moved to be utilized in a different environment, the information gain could be transferred to the set of sensors in the new environment from the old one appropriately. Another example could be if a sensor in a dynamic setting stops streaming data, but another sensor measuring the same property starts streaming at a later point in time. In this case the information gain connected to the sensor which disappeared can be transferred to the new sensor. In both cases information that has been accumulated can be reused through transferring the information gain.

In Active Learning it is generally assumed that the “oracle” will always answer a query and that this answer is correct. In reality this might not be the case. A DIVS that is operating in an office environment, for instance, will most likely direct its queries towards the people currently present in the environment. However, the users providing feedback can change over time, which in turn will affect how the data is labelled in several ways. Firstly, different people might label the same state in different ways. For instance, a state labelled by one user as ‘meeting’, can be labelled by another user as ‘presentation’. Secondly, the reliability of the answer from a user may differ.

## 5. Conclusions

In this paper a refined version of the Dynamic Intelligent Virtual Sensors (DIVS) concept was introduced in order to support the provision of applications based on sensor data. The ambition of the concept is to address a number of challenges identified by considering real word applications together with companies. The concept includes a minimal representation of the DIVS output to address the challenges identified in a dynamic environment with respect to sensor availability. Some properties of the concept were validated through experiments including an interactive machine learning mechanism. In particular, different strategies for making use of labelled data from potential users were studied. It was shown that the distribution of different budget allocations for labelled data and the approach of a proactive user for labelling had a rather important impact on the accuracy.

The results suggest that a relatively good accuracy can be achieved despite a limited budget. Further improvement of the performance can be achieved by proactive labelling, that is to provide correct labels when the output is wrong. Interestingly, the results indicate a significant deterioration of performance can be obtained if too much of the budget is used up early in a real-world environment where sensor may disappear. Hence, it is evident that the careful selection of how to make use of labels provided by users in an environment with dynamic sensor availability is very important. This indicates that designing applications where users provide feedback when the system output is incorrect, is more efficient than having users regularly provide correct labels. Also, by using a learning strategy for managing the budget of labels combined with a pro-active user behavior, the need of user feedback can be further reduced.

## Figures and Tables

**Figure 1 sensors-19-00477-f001:**
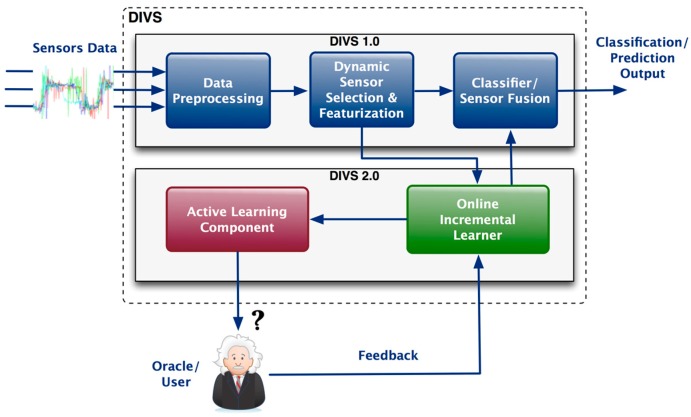
The DIVS data processing pipeline. DIVS 1.0 belongs to previous work and DIVS 2.0 is the extension included in this paper.

**Figure 2 sensors-19-00477-f002:**
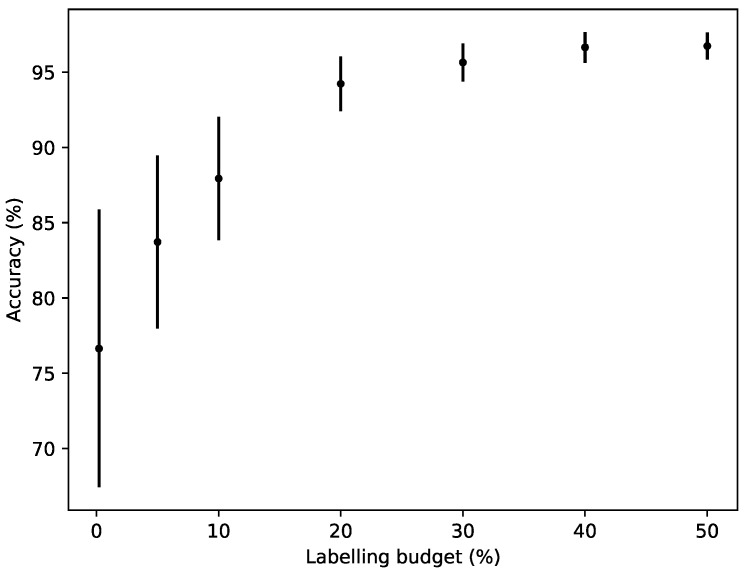
The accumulated accuracy for different labelling budgets.

**Figure 3 sensors-19-00477-f003:**
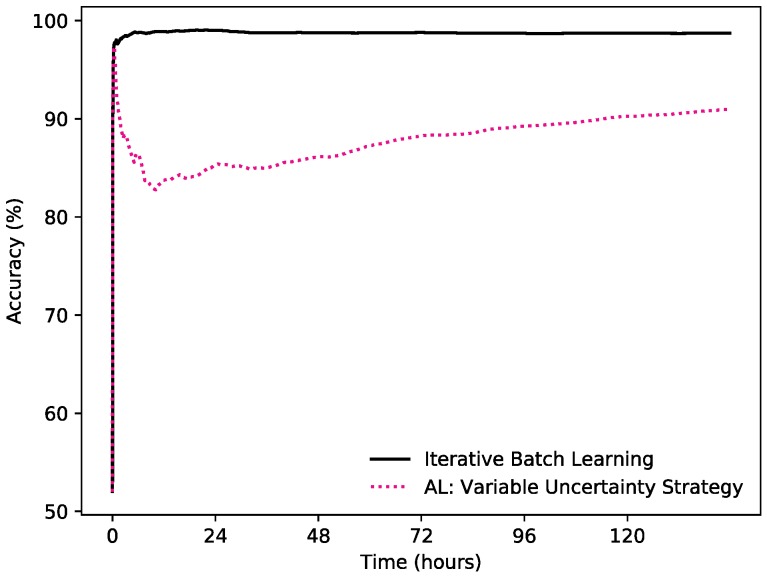
The accumulated accuracy for iterative batch learning vs. variable uncertainty strategy.

**Figure 4 sensors-19-00477-f004:**
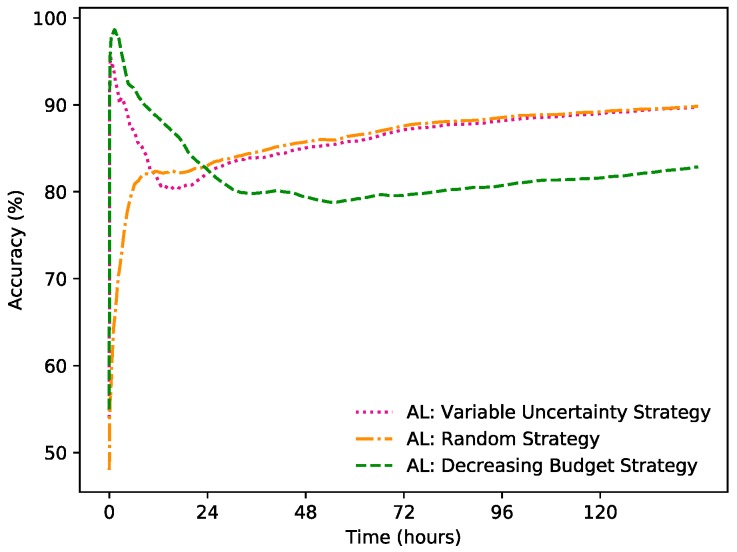
The accumulated accuracy for Variable Uncertainty Strategy vs. Random Strategy vs. Decreasing Budget Strategy.

**Figure 5 sensors-19-00477-f005:**
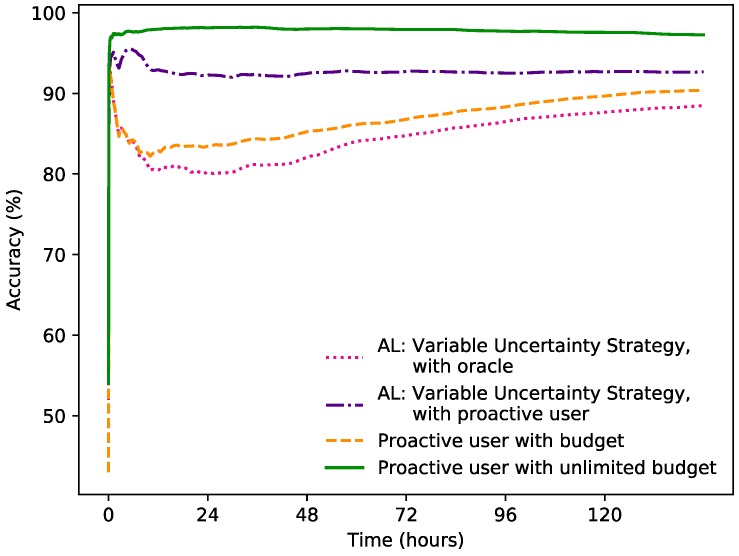
The accumulated accuracy for Variable Uncertainty Strategy with and without proactive user vs. Proactive User with and without budget limit.

**Table 1 sensors-19-00477-t001:** The accumulated accuracy and F1-score for learning strategies in static settings.

Learning Strategy	Accumulated Accuracy, after 1 h	Accumulated Accuracy, after 6 h	Accumulated Accuracy, after 144 h	F1 Score, after 144 h
Iterative Batch Learning	97.80%	98.79%	98.71%	0.9698
Variable Uncertainty Strategy	92.39%	86.33%	90.94%	0.8111

**Table 2 sensors-19-00477-t002:** The accumulated accuracy and F1-score for learning strategies in dynamic settings.

Learning Strategy	Accumulated Accuracy, after 1 h	Accumulated Accuracy, after 6 h	Accumulated Accuracy, after 144 h	F1 score, after 144 h
Variable Uncertainty Strategy	94.38%	87.01%	89.73%	0.7645
Random Strategy	63.92%	80.67%	89.84%	0.7589
Decreasing Budget Strategy	98.39%	91.90%	82.87%	0.6190

**Table 3 sensors-19-00477-t003:** The accumulated accuracy and F1-score for user strategies in dynamic settings.

Learning Strategy	Accumulated Accuracy, after 1 h	Accumulated Accuracy, after 6 h	Accumulated Accuracy, after 144 h	F1-Score, after 144 h
Variable Uncertainty Strategy, with oracle	89.95%	83.99%	88.49%	0.7442
Variable Uncertainty Strategy, with proactive user	94.92%	95.35%	92.69%	0.8144
Proactive user with budget	89.80%	84.18%	90.43%	0.7783
Proactive user with unlimited budget	97.26%	97.60%	97.29%	0.9329

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
