# Peer review of "Collaborative Sensing with Interactive Learning using Dynamic Intelligent Virtual Sensors"

_sensors, 2019, doi:10.3390/s19030477_

Round 1

Reviewer 1 Report

The paper explain a very interesting research about virtual sensors. In particular, the DIVS, that are an advanced and mixed sensoring to create improved virtual sensors.

I think that the paper is well structure and well explained; however, these are the point that I think that the authors must correct to improved the quality of the paper:

- In section 2, the DIVS concept are explain, but I miss the explanation about the "Classifier/Sensor Fusion" block in Figure 1. At the end of this section, all the blocks in the figure are described, but not the "Classifier/Sensor Fusion".

- Figure 2 in Result section, should have the "x" axis in %, in the same way that the "y" axis. Moreover, in the explanation of this figure, apear a 0,2% labelling budget, that I really don't know what it means.

- Figures 4 and 5 should have different types of lines for the different variables; it helps the reader to identify the correct line in blakc and white print.

Author Response

- In section 2, the DIVS concept are explain, but I miss the explanation about the "Classifier/Sensor Fusion" block in Figure 1. At the end of this section, all the blocks in the figure are described, but not the "Classifier/Sensor Fusion".

We have added text describing some about that block in 2.3 “online learning” where we also refer to previous work.

- Figure 2 in Result section, should have the "x" axis in %, in the same way that the "y" axis.

This has been updated.

 Moreover, in the explanation of this figure, appear a 0,2% labelling budget, that I really don't know what it means.

We have included a explanation.

- Figures 4 and 5 should have different types of lines for the different variables; it helps the reader to identify the correct line in blakc and white print.

This issue with the figures has been addressed.

General comment:

We have also updated the Figure 1 with respect to the “oracle” due to ensure fair use of copyrights

Reviewer 2 Report

This paper is an extension to a previous work. It is a refined version of the DIVS concept by integrating an interactive machine learning mechanism using Naive Bayes classifier and the variable uncertainity strategy for active learning.

Here are some notes to consider:

Figure 1 was mentioned in page 4 and the figure itseld is on page 6.

the components of the architecture (Figure 1) were discussed except Data pre-processing, pre-processing was mentioned slightly in page 6. It was mentioned that data pre-processing includess smoothing, identifying and removing outliers and resolve any issue about data consistency. There is no explanation how these pre-procesing procedures will be excuded.

Figure 2 was inserted on top on page 9 before refering to it in the context later. The same for figure 5. Please check all the figures and tables and where is was refered to.

Author Response

Figure 1 was mentioned in page 4 and the figure itseld is on page 6.

The position of Figure 1 has been changed

the components of the architecture (Figure 1) were discussed except Data pre-processing, pre-processing was mentioned slightly in page 6. It was mentioned that data pre-processing includess smoothing, identifying and removing outliers and resolve any issue about data consistency. There is no explanation how these pre-procesing procedures will be excuded

We did not produce the dataset ourself but is assumed to be carried out by the authors of the dataset if needed (although not explicit explained in the source).

Figure 2 was inserted on top on page 9 before referring to it in the context later. The same for figure 5. Please check all the figures and tables and where is was referred to.

The position of all figures and tables has been checked in regards to where they are mentioned in the text and changed if needed.

A general comment:

We have  updated  Figure 1 with respect to the “oracle” due to ensure fair use of copyrights